# Practical Opportunities to Improve the Impact of Health Risk Assessment on Environmental and Public Health Decisions

**DOI:** 10.3390/ijerph19074200

**Published:** 2022-04-01

**Authors:** Tine Bizjak, Davor Kontić, Branko Kontić

**Affiliations:** 1Department of Environmental Sciences, Jožef Stefan Institute, Jamova cesta 39, 1000 Ljubljana, Slovenia; davor.kontic@ijs.si; 2Jožef Stefan International Postgraduate School, Jamova cesta 39, 1000 Ljubljana, Slovenia; branko.kontic@ijs.si

**Keywords:** risk analysis, health risk assessment, health impact assessment, risk management, decision analysis, public health

## Abstract

Following alerts about the diminishing role of health risk assessment (HRA) in informing public health decisions, this study examines specific HRA topics with the aim of identifying possible solutions for addressing this compelling situation. The study administered a survey among different groups of stakeholders involved in HRA or decision-making, or both. The responses show various understandings of HRA in the decision-making context—including confusion with the health impact assessment (HIA)—and confirm recurring foundational issues within the risk analysis field that contribute to the growth of inconsistency in the HRA praxis. This inconsistency lowers the effectiveness of HRA to perform its primary purpose of informing public health decisions. Opportunities for improving this situation come at the beginning of the assessment process, where greater attention should be given to defining the assessment and decision-making contexts. Both must reflect the concerns and expectations of the stakeholders regarding the needs and purpose of an HRA on one side, and the methodological and procedural topics relevant for the decision case at hand on the other. The HRA process should end with a decision follow-up step with targeted auditing and the participation of stakeholders to measure its success.

## 1. Introduction

Recent research has recognized a continuous spread of fundamental issues in health risk assessment (HRA), as well as a poor, or at least unclear, link between HRA results and (risk management) decision-making [1,2,3,4,5]. Some studies expressed concern about inconsistent practices that are drifting away from the definition and generally approved process of HRA [6,7,8,9], while others pointed out the possible detrimental effects of a narrow understanding of HRA founded on the deleterious interchange of HRA with hazard assessment, which results from an unclear understanding of the differences between hazard and risk [10,11]. In general, the term “risk” is used too loosely too often. Considering the basic meaning of “risk” [12] and how the term is understood in the risk analysis (RA) discipline [13], the term is regularly interchanged with terms such as “hazard”, “possibility”, “potential”, “threat”, and “safety”.

The inconsistent or improper use of “risk” in everyday talk, by media, semi-scientific literature, etc., can contribute to the uncritical spread of a superficial understanding of the risk concepts, which might lead to a loss of the effectiveness of HRA for informing environmental and public health policy decisions and could negatively affect their implementation and public compliance [14]. This is what we observed during the COVID-19 pandemic on a daily basis [1,3]. For instance, phrases such as “risk of exposure”, “risk of going out”, and “risk of partying” [15,16,17] were being popularly used with the effect of pushing off the risk scientific more consistent ones [13], such as “risk of mortality in patients infected with SARS-CoV-2” [18], and without paying attention to the differences in their meanings. Notwithstanding their description as science based [19], the measures adopted throughout the world during the current pandemic often lacked clear and systematic support and justification by the results of targeted HRA or other public health impact evaluations but seemed unclear, more intuitive, non-transparent, and political, and in many cases imposed [20,21,22]. A better level of underlying understanding of RA terms and principles [19] during the COVID-19 pandemic could contribute to easier implementation and better compliance with even the more drastic measures, such as public life closures and travel restrictions [23].

### 1.1. Effectiveness of HRA for Decision-Making

A prerequisite for a good public health policy is building upon the experiences of past or existing policies, which requires well-planned and systematic follow-up activities that monitor policy implementation and its impacts [24,25]. Policy decision-making challenges arise when knowledge about health risks or impacts related to specific activities and exposures is uncertain or incomplete. The field of HRA was developed to address knowledge gaps and mitigate them [26]. In general, HRAs intend to support public policy decisions, set priorities among research needs, and help develop ways to evaluate the costs and benefits of regulatory decisions [7,9,27,28]. Challenges for policy decision-making, e.g., related to knowledge gaps about different health risks associated to hazardous substances, are observed throughout the decades. An example is the regulatory process concerning glyphosate in the EU [29]. Its use as an active substance was approved until 15 December 2022, with pending regulatory decisions regarding the renewal of approval [30]. There is no consensus regarding the adverse health effects related to glyphosate exposure [31,32]. The International Agency of Research on Cancer classified glyphosate as “probably carcinogenic to humans” (Group 2A) in 2015 [33], while both European Chemicals Agency and the United States Environmental Protection Agency concluded that glyphosate does not meet the criteria for classification as a carcinogen [34,35,36]. Despite several lawsuits about glyphosate exposure-related adverse health effects, awarding multi million dollars damages to customers in the US, the enduring belief in the safety of Roundup (i.e., glyphosate-containing weed and grass killer) by its manufacturer continues to be unwavering [37].

The practice of “risk assessment of chemicals”, where a mere comparison with selected reference values is misused to represent or characterize actual health risks [38], has become common in the research area [39,40,41] and its application for administrative and regulatory purposes is even required [42]. Generic regulatory approaches to risk assessment, as adopted by the REACH legislation [42], have strayed from the basic HRA concepts [9] by focusing primarily on the intrinsic properties of substances (i.e., hazard identification) and less on exposure assessment [43]. Failing to reflect actual exposures in specific settings (e.g., working conditions) [44], such assessments can lead to misleading or wrong conclusions about safety.

Factors contributing to the effective consideration of HRA findings in public health policy remain understudied and insufficiently understood [45,46]. The multitude of different risk assessment processes, for example, as applied in practice by the US federal government, makes it difficult to understand the relation of risk assessment to environmental protection, environmental health policy, occupational health, and regulation [47]. In this context, it is important to expose continuous international efforts coordinated by the International Atomic Energy Agency (IAEA) and OECD Nuclear Energy Agency (NEA) to link basic concepts, standards, methods, tools, and best practice in the area of risk assessment and risk-informed decision-making [48,49,50,51]. Efforts to consolidate the RA area [52] are often ignored by “self-styled risk analysts” [53]. The authors have experienced this recently on multiple occasions during their involvement in several EU-funded projects that dealt with HRA in the context of making improvements to public health policies or guiding decisions for reduced exposures to hazardous substances, such as HBM4EU (https://cordis.europa.eu/project/id/733032, accessed 29 March 2022), HERA (https://cordis.europa.eu/project/id/825417, accessed 29 March 2022), ICARUS (https://cordis.europa.eu/project/id/690105, accessed 29 March 2022), and NEUROSOME (https://cordis.europa.eu/project/id/766251, accessed 29 March 2022). It remains to be seen whether the European partnership for the assessment of risk from chemicals (PARC) [54] will succeed in changing the REACH legislation’s “risk of chemicals” approach toward returning the HRA methodology to its origins [9].

In summary, the lack of understanding of the HRA process and its application causes HRA and risk management frameworks to oversimplify complex scientific assessments, resulting in misunderstandings and hindering risk management decision-making, its success, and confidence in it [2,53]. Constant development of HRA practice is necessary to keep up with the advancing knowledge of relevant scientific fields [55,56]. However, advances in specific disciplines with potential usefulness for specific HRA elements (e.g., use of human biomonitoring [57], modeling, in vitro, and in vivo/in silico exposure/dose-effect/response studies [55]) alone cannot ensure improvements in HRA’s effectiveness in informing public health decisions without the wide recognition and consolidation of fundamental HRA principles and concepts.

### 1.2. Brief Summary of Issues with a Research Look Ahead

Table 1 provides an overview of fundamental HRA concepts and related issues assumed to influence the basic understanding, i.e., the theory of HRA on one side and the practice, specifically the effectiveness of HRA results in informing risk management decisions, on the other. The table does not provide a complete list of issues in the RA area but focuses on those assumed to influence the impact of HRA in informing public health decision-making. We believe that these issues deserve full research and policy consideration, with proper intervention to secure HRA from further conceptual erosion.

The research behind the above-described challenges is mostly descriptive (i.e., observational) and as such does not include specific attempts, guidance, or proposals for searching solutions, neither do any other accessible publications deal with concrete ways of stopping the further detrition of fundamental issues and the spread of inconsistencies within HRA. It seems, therefore, that targeted intervention studies, which are common in the area of community-based participatory research (CBPR) and especially in CBPR public health that lead to improvements [58,59], should be performed in the HRA area as well.

The study presented here has both observational and interventional elements and is pioneering in its call for thorough consideration and the re-establishment of an overall understanding of the effectiveness of HRA for decision-making. Its interventional contribution is provided in the form of proposals and recommendations, as given in the following section.

## 2. Materials and Methods

In consideration of the broader concepts presented in Table 1 we derived the following three more focused assumptions as a basis for our study and for designing the two questionnaires used in the survey:The informing potential of HRA results is limited because some of the various types of results may not conform to or properly fit the area/policy of their application.HRA is not applied in a consistent and integrated manner; rather, only some elements of HRA are practiced because of a limited understanding of the overall process of HRA, particularly its purpose.There are diverse understandings of the importance of different elements of HRA for public health decision-making. This is evident from the interpretation of HRA results, especially in cases when consultation with the users of HRA results is poor or is missing, so the interpretation is biased by, for example, assessors, or in the opposite case, when the users of the results are consulted; however, a deeper exploration of their different understandings is missing.

To evaluate the validity of the assumptions, we administered a survey to different stakeholders to ascertain their understanding of fundamental HRA concepts and principles. We used their responses to evaluate the distribution of inconsistencies and inadequacies in their understanding of selected topics in the decision-making context. These findings formed the basis for developing proposals and recommendations to improve the impact of HRA on environmental and public health decisions.

The targeted stakeholders were four carefully selected groups of professionals from various backgrounds (i.e., research, administration, public health, and economy) (Table 2). Before administering the survey, the stakeholders were checked for their involvement in the RA area based on their expertise, previous work, or interests. Because of the multidisciplinary nature of professionals involved in RA, the limited availability of resources, the anonymity of the collected responses, and potential questionnaire biases, we restricted our sample to professionals who were willing to respond to the survey and who were reasonably easily accessible, for example, involved in the same projects or other activities as the authors and with publicly accessible contact information.

The first group comprised participants in a workshop organized by the Slovenian Research Agency-funded project. The participants were from the industry, the Department of Environmental Sciences at Jožef Stefan Institute, Chemicals Office of Republic of Slovenia, Slovenian National Institute of Public Health, and the Faculty of Health Sciences of the University of Ljubljana. Several follow-up interviews with the respondents were conducted to clarify the questions and ambiguous responses. The group was relatively heterogeneous in terms of the participants’ backgrounds, fields of work/expertise, and usage of HRA results for their specific involvement in decisions, and included well-experienced professionals. The second group comprised early-stage researchers (ESRs) and other researchers involved in the NEUROSOME project. ESRs were selected and asked to participate in the survey since their knowledge of HRA and perceptions of relations between HRA results and decision-making were still under development. The distribution of their answers to the questionnaire was expected to show some inconsistencies, which, after comparison with the responses of the first group, could be attributed to their inexperience in both HRA practice and involvement in decisions. The third group comprised participants in Environmental Health Risk: Analysis and Applications educational activities organized by Harvard T.H. Chan School of Public Health in 2020. The participants were professionals from regulatory agencies, such as the United States Environmental Protection Agency and the United States Food and Drug Administration, from universities, private consultant companies, and industries in the US, Europe, and Asia. The fourth group comprised established RA and decision analysis professionals, including notable members of the Society for Risk Analysis (SRA), and authors of prominent publications in the areas of RA and decision analysis. This group acted as a reference group in the context of the survey (see more in the Discussion section). Invitations to respond to the questionnaire were distributed by email and through internal SRA pages.

We distributed the same 15-item questionnaire in English or Slovenian to the first three groups and a more focused seven-item questionnaire to the fourth group of professionals, either through email or by sharing the link to the questionnaire on Google Forms. The aim of the more focused questionnaire, which was modified based on responses from the first three groups, was to provide more transparent and justifiable stand-points of the esteemed experts regarding the key issues about HRA in the decision-making context. Besides comparison of the understanding of selected topics with the professionals from the other three groups these were expected to act as a guidance and additional argumentation for recommending ways of improving the impact of HRA on decision-making. This prevailing focus of the second questionnaire had a consequence of leaving out questions about health impact assessment (HIA) and differences between HIA and HRA, which were included in the first questionnaire. The responses from all groups were collected anonymously or were anonymized before the analysis.

The survey was non-probabilistic. It was judgmental—all four groups were carefully targeted as mentioned above. Non-probabilistic type of sampling meant that the use of probabilistic statistical tests (for which conditions were not met) are not applicable. Consequently, the statistical analysis included calculations of proportions, mean values, and standard deviations, which were intended for qualitative relative comparisons, such as between the four groups or in relation to the responders’ background. The responses with three-point Likert scales (first three groups) were assigned values of 1, 3, and 5 for a more straightforward comparisons of their responses with the responses from the fourth group that used a five-point Likert scale. This should be regarded with caution when interpreting results, and should only be done when comparing the relative differences between the options and not for comparing their absolute value, since it increased the apparent gap between the absolute values of different responses. Open-ended (i.e., other) options in the multiple-choice questions were intended for comments or explanations and not to restrict the responses only to the options specified. Since the responses to open-ended questions served as a source of additional information, their detailed analysis did not seem practical and beneficial.

The Appendix A includes both questionnaires and a summary of the responses.

## 3. Results

The responses confirmed our first assumption that the informing potential of HRA is limited because some of the various types of results may not conform to or do not properly fit the area/policy of their application. Responders understood HRA as being important for decision-making, but did not show a consistent understanding of how HRA should influence decisions. More than two-thirds (68%) of the respondents in the first three groups answered that HRA improves the transparency of the decision-making process, and 55% regarded HRA results as direct and the most important basis for decision-making. Similarly, 67% of the respondents in the fourth group regarded HRA and its results as important in influencing a wide range of decision-making considerations. However, respondents in the fourth group showed a coherent understanding that HRA results are not the direct and most important foundation for decision-making. In answers to open-ended options, they pointed out a decision context, i.e., concrete situation, as well as needs of the decision-makers which determine appropriate types of HRA results or endpoints.

The question about the most useful types of HRA results for decision-making aimed to assess whether there are specific types of HRA results that are generally preferred The HRA results expressed probabilistically were understood as being only slightly more useful in decision-making (Figure 1). As in the first three groups, no clearly preferred type of HRA results that would be the most useful in informing decision-making was identified in the responses of the fourth group. The responses from group 3 were more similar to the responses from group 4 compared to the responses from groups 1 and 2. “Other” types of useful HRA results were only specified by the fourth group. This group again stressed the importance of a decision context for determining most useful HRA results. Analysis of responses from the first three groups in relation to the respondent’s background showed only small differences (Appendix A). Next, we attempted to identify whether specific types of HRA results were preferred in different decision-making settings, such as the economy, public health policy, and health protection actions. Only small differences in the types of HRA results that were understood as useful in these areas were observed. Types of HRA results that included costs were understood as being slightly more useful for the economy.

In addition to the questionnaire responses, the interviews and discussions were performed during the survey within the first three groups. These showed that responders who claim to perform HRAs often follow a certain type of HRA procedure and select a certain type of HRA results (e.g., comparison with reference values, standards, guidance values) without being actively involved in a decision-making process from the beginning until the end. HRAs performed in this way can be lacking in understanding and consideration of expectations regarding the HRA results in the decision-making context. If so, this raises two fundamental issues. One is about assessment context (what is to be assessed, how to be assessed and why), which may not be considered thoroughly. The other is that a situation of “throwing results over the fence” in the RA arena [60] may occur, where risk assessors expect that their results will be used by others without any consultation, collaboration, and checking the “fitness for purpose” principle.

Our second assumption that HRA has not been applied in a consistent and integrated manner, and that only some elements of HRA have been practiced, was confirmed by responses about HRAs based on comparisons with standards, guidance values, reference values (e.g., hazard index or quotient), or tolerable daily intake values, and by a lack of understanding about the similarities and differences between HRA and HIA. Almost all respondents in the first three groups (93%) understood comparisons with standards, guidance values, or reference values as important in interpreting HRA results. This indicates that even in case when HRA is not performed in a consistent and integrated manner, the results of comparison with reference values are perceived important in interpreting the results. Such understanding, similar as mentioned above, exposes the issue of weak consideration of the assessment context. However, none of the established professionals from the fourth group selected hazard indexes or quotients among the most useful types of assessment endpoints in decision-making. The majority of the respondents (55%) in the first three groups answered that comparisons with guidance values improve decision-making, but the meaning of violation of guidance values is not clearly understood. About one-third (38%) of the respondents in the first three groups answered that the violations of guidance values should lead to the prohibition of the substances/activities causing violation, and one-third answered that it is not clear what actions should be taken. Less than half of the respondents in the fourth group answered that such comparisons are important for informing risk communication regarding decision-making, with one-third answering that such comparisons are not necessary and may mislead decisions toward radical attitudes.

The third assumption that there are diverse understandings of the importance of different elements of HRA for public health decision-making was also confirmed by inconsistent and dispersed responses. Elements such as HRA procedure, participants in the assessment process, reasons for conducting HRAs, and the reputation and credibility of the risk assessor were understood as additional important types of information in the decision-making, but were understood as relatively less important compared to the importance of other HRA elements. The survey respondents in the first three groups understood all HRA elements as being very important or of medium importance. Relatively less importance was ascribed to uncertainty, coordination of HRA procedures, and the type of HRA results, regardless of the responders’ backgrounds (Appendix A). The responses of the fourth group showed that decision alternatives for mitigating exposure, uncertainty of HRA results, and transparency and clarity of the assessment process—the latter, contrary to perception of the first three groups, emphasizes proper consideration of the assessment context—were evaluated as more useful in informing decision-making than other elements.

A detailed summary of all responses from the four groups is included in the Appendix A.

## 4. Discussion

### 4.1. Comments on Survey Responses

While respondents considered the influence of HRA on decision-making important, it can be concluded that the perception of its importance may be biased. This bias is clearly evident from the responses that evaluated the importance and impact of HRA in decision-making based on standards and reference values. According to the observed perception, HRA should be done as a comparison with reference values, which, if viewed from the risk management perspective, suggests that decision-making without reference values is not possible. This perception is inappropriate and is not in accordance with the generally accepted theory, definitions, and processes of HRA [9]. Many responders in the first three groups either did not clearly understand the core RA subjects [52] or did not respond consistently, which supports the recognized need to consolidate the fundamental principles and core subjects of RA among relevant scientific and nonscientific communities [1,3]. This consolidation can be done in a variety of ways, such as through educational activities and university programs and workshops. Consolidation is also needed with regard to the overall process of HRA.

No clearly preferred type of HRA results was identified that would reflect a consistent understanding of the main RA concepts and would acknowledge probability and severity as key determinants of risk [1]. Threshold-based types of HRA results (i.e., comparisons with guidance values) are promoted by authorities and regulatory bodies [61,62], and are often reported as the only measure of risks [63,64], even though they are actually arbitrary measures of concern [65]. Furthermore, guidance values are not only science-based but are defined in an administrative process that determines acceptable risk and considers scientific uncertainty, risk management options, economic benefits and costs, relevant laws, and social norms [62]. Underlying characteristics, including the strength of the knowledge applied in setting each guidance value, and clear evaluation of the case-specific applicability of selected guidance value-based HRA, are often not transparently reported or considered in publications [63,66]. Such practice blurs the delineation between science and judgments [2] and encourages the abandonment of the fundamental exposure–response concept of HRA [67]. There is too much focus on the simple fact of exposure (e.g., contact with hazardous material) rather than the way (situation), duration, mode, and amount of exposure [53] with epidemiological data, and a lack of understanding that regardless of the existence or nonexistence of guidance values, even modest exposure reductions can benefit public health [67].

The types of HRA results incorporating probability were identified as more useful in decision-making (Figure 1), while the uncertainty was not consistently understood among the more important elements of HRA (Appendix A), indicating a lack of acknowledgment that uncertainty and variability are inherent characteristics of HRA [27,68]. By contrast, the responses of the fourth group showed better consistency with basic RA concepts (Appendix A). Since probability is often difficult to communicate, it does not influence decision-making and decisions as it should. An inadequate understanding of probability can lead to its reduced or discarded consideration in favor of the magnitude of risk, which leads to cognitive biases [69] and poor decision-making. When certain elements of HRA are less known or understood, biases seem to point to the area of the analyst’s expertise [70].

### 4.2. Opportunities for Consolidating Understanding and Improving the Utility of HRA

Besides general recommendations, such as more focused education opportunities or academic discussions [3,71], there is a lack of concrete recommendations on how to improve the situation related to a particular HRA issue or to the HRA process as such. There is also a lack of targeted, intervention studies that would provide a solid background for concrete improvements. In this view, our study could be seen as an attempt to fill this gap.

An analysis of the survey showed that there are no clear patterns or causes for the inconsistent, dispersed, and diverse understanding of HRA. Confusion between HRA and HIA further supports the conclusion that there is a lack of consistent understanding about why and when to perform specific assessments, how they fit in the context of specific public health problems and decisions, and how they can meet the expectations of the decision-makers and other stakeholders. Dispersed inconsistencies with no clear patterns or causes limit the potential of targeted efforts aiming to improve the understanding of HRA in decision-making. Therefore, instead of focusing on particular or only some HRA elements, we address the observed inconsistencies and foundational challenges in the RA discipline [1], such as those presented in Table 1 by highlighting opportunities for procedural improvements of HRA praxis (Figure 2) that aim to encourage and ensure overall improvements in the effectiveness of HRA in environmental and public health decision-making. The concepts behind the framework in Figure 2 are consistent with other recognized HRA frameworks [7,9,27,55]. The framework does not aim to replace them and should not be considered an all-inclusive guide on how to perform HRA. Its purpose is more focused—to highlight how the risk-informing potential of HRA can be clarified and improved.

Apart from a few open-ended i.e., “other” responses to several questions (e.g., that the selection of the assessment endpoints or the importance of other important information depends on the assessment or decision context), most of the survey responses indicated a lack of understanding of the fit-for-purpose concept of HRA [7]. It is not reasonable, or even wrong, to assume the credibility or relevance of the specific findings of an HRA without assuring that the assessment is fit for its intended purpose. Fitness for purpose is crucial for improving the utility and effectiveness of HRA in informing decisions and should be ensured from the earliest stages of the assessment process. The origins of every HRA are in the initiation phase, which should gain full attention from risk analysts and assessors. It should be recognized, however, that the initiation phase requires a lot of consultations, tolerability, respect, and patience among the stakeholders involved, since there are often unclear and vague or unspecific expressions and issues that need to be clarified, consolidated, and eventually approved (blurred cloud in Figure 2). Such issues include perceived health concerns in different population groups. Therefore, the initiation phase must support the active participation of stakeholders. This can be facilitated by decision analysis tools and methods [72,73] that improve the understanding of stakeholders’ concerns and values, help identify and clarify the actual decision problem and its most important components, and determine whether and how detailed assessments of health risks are actually needed and how they should be performed [73,74,75,76].

“It is impossible to create meaning about risks without defining the context in which it is created” [77] (p. 4). Systematic, consolidated, and consistent discussions during the initiation phase are expected to generate the first formal step of HRA, which must clarify the assessment context (Figure 2, step 1) [78]. The assessment context reinstates the importance of planning, scoping, and problem formulation [7], which have not yet been acknowledged sufficiently as a regular and necessary step in HRA practice [56,63], as also shown by our survey. It clarifies the understanding and expectations regarding the assessment process, the results of HRA, and the context of their use among all involved in the assessment process before more technical steps of the assessment are conducted [27]. The broader context of a decision problem should be distinguished from the more specific context of the assessment. Decision analysis tools and methods are proposed to address the practical challenges related to complex processes of planning, scoping, and problem formulation. Clarification is especially needed in situations with diametrically opposed interests of stakeholders and where different options for solving a problem at hand are considered. The assessment context should be linked to the context of the decision problem and should clarify if the HRA is needed for addressing the decision problem and in what scope. There are cases and decision contexts which do not require full HRA and where only exposure or hazard assessment suffice decision needs. After establishing a clear need for HRA, the assessment context should determine the purpose of the HRA by clarifying what is going to be assessed, why it is to be assessed, the assessment’s scope and plan (methods, tools, staff, financing, duration, etc.), and the decisions to be supported by the results of the HRA [27,79]. The concepts behind the assessment context step are consistent with the recently published recommendations for decision-first modeling for emerging risks [80], with an emphasis on a continuous and much broader inclusion of stakeholders, e.g., risk assessors, relevant subject matter experts, decision-makers at different administration levels, those affected by health risks, concerned citizens, and non-governmental organizations (NGOs), which is to ensure that the results of HRA are effectively addressing the concerns of as many stakeholders as reasonable.

Only after the assessment context is clear should the assessment process continue with “classical” HRA steps, including the four that often receive too much attention without a clear link to expected decisions [81]: hazard identification, dose-response assessment, exposure assessment, and risk characterization (Figure 2, step 2). These four steps, described in detail elsewhere [7,61,62], should be performed in consideration of the previously clarified assessment context. Based on the availability or non-availability of information required for specific estimates (e.g., exposure estimates, dose/exposure–response relationship), a need may arise for re-evaluations and modifications of the assessment context, considering new information coming from most recent relevant studies or decision needs. The continuous participation of stakeholders, whose importance was underacknowledged, improves the understanding of health risks, as well as the basic concepts of risk, hazard, and probability, which are often confused with each other [53]. Their participation can also address the issue of the often limited amount of information about assumptions and uncertainty factors in numerical estimates of hazards and risks received by risk managers and stakeholders [82].

To improve the understanding of the value of HRA for public health decisions and its contribution to reducing undesired health outcomes, it is necessary to monitor the decisions and their implementation with various follow-up post-decision evaluations, such as monitoring and auditing [83,84] (Figure 2, step 4). Follow-up evaluations should evaluate the success of HRA—for example, if objectives set in the assessment context are met—and must utilize tools and measures that are compatible with those used in HRA [24,85,86]. Such evaluations should also assess all relevant technological advances that could contribute to additional exposure reductions and are relevant in decision-making on various levels [24,25]. The findings of follow-up evaluations must be subjected to stakeholders’ scrutiny, which contributes to building confidence in the overall assessment and decision process, provides trust and respect among the parties involved, and could eventually be a factor in the initiation stage of a fresh HRA process.

Proposed opportunities for improving the utility of HRA for decision-making are applicable generally, despite being based on authors’ experience in HRA and HIA in the area of non-infectious diseases caused by activities leading to exposure to hazardous substances. Additional information about improvement opportunities within the HRA framework is provided in the Appendix A, which also includes additional information that supports our recommendations.

### 4.3. Comments on Similarities and Differences between HIA and HRA

We observed a poor basic understanding of HIA and HRA and of the differences between them. As a response, we list some of the key differences between HRA and HIA as guidance for reaching consistency on the topic:WHO’s Gothenburg consensus paper defines HIA as “a combination of procedures, methods and tools by which a policy, program or project (i.e., a development proposal) may be judged as to its potential effects on the health of a population, and the distribution of those effects within the population” [87] (p. 4). In this context, the HIA is part of a formal procedure, either an environmental impact assessment (EIA) or a strategic environmental assessment (SEA), which are required and determined by the EIA and SEA directives, respectively. The role of the HIA is to consider whether a development proposal could be improved in terms of protecting public health [88]. Methods and tools applied in HIA include expert opinion, historical data application, interaction matrices, scenario analyses, and other desk studies. Specific measurements and epidemiological studies are usually beyond the scope of HIA due to time constraints and limited financing. However, if applicable study results exist (including possible HRA results), they could be used.HRA “is the process to estimate the nature and probability of adverse health effects in humans who may be exposed to chemicals in contaminated environmental media, now or in the future” [89] (first paragraph). Methods and tools applied in HRA include laboratory experiments, comprehensive modeling, specific measurements, and epidemiological studies.HIA is triggered by a new development proposal [90], while an HRA can be initiated in response to a specific health concern by citizens, researchers, public health professionals, NGOs, administrators, etc.Despite some similarities, HIA and HRA should be distinguished [91]. The inconsistent use of HIA and HRA terminology creates confusion not only among the scientific community but also among potential users of the results of an HRA or HIA. While HRA provides a set of types of results, as we discuss in this paper, HIA’s ultimate results are in the form of recommendations for changing/improving a development proposal to cause the least health issues during its implementation and subsequent utilization.HRA can be a part of HIA but not vice versa.

### 4.4. Recommendations

The following is a summary of the recommendations:A careful, technically sound narration in HRA should always be applied. The term ‘risk’ applies to the quantitative, probabilistic expression of the occurrence of specific health damage as a consequence of a certain exposure. Its interchange with the term “hazard” [13] is to be strictly omitted. Vague, unclear, populistic, and otherwise non-specified and non-quantified usage of the term ‘risk’ is neither appropriate nor valid. The adequacy of other derived expressions depends on the area of application, e.g., in the economy—probable additional costs due to workers’ absenteeism associated with health consequences as assessed in the specific HRA.Comparative evaluations using reference or guidance values of environmental pollution, human biomonitoring data, exposure, etc., could be used to evaluate the level of health concerns (e.g., for prioritization purposes). However, these semi-quantitative and qualitative evaluations should not be recognized, whatsoever, as a limited HRA or as actually characterizing specific risks.Any HRA should start with a clarification of the decision and assessment contexts, which must guide all subsequent steps of the assessment. The decision adopted and implemented based on HRA results should be monitored for the success and expectations of the concerned stakeholders.All HRA steps, as shown in Figure 2 or other frameworks, Ref. [7], should be practiced (and not only the four “classical” HRA steps). Types of HRA results should follow measures and indicators applied in epidemiological studies (i.e., addressing actual health concerns in a population of interest). Consideration of the results of relevant epidemiological studies is inevitable during the quantification and characterization of health risks (in environmental and public health areas).Inconsistent, free narrative use of “health risk assessment” or “health impact assessment” phrases, contrary to their established meanings, procedures, and differences, brings additional confusion in the area [92,93,94] and should be avoided.HRA is not HIA, and HIA is not HRA; however, HIA can involve HRA results. A distinction between the two is crucial for consolidating HRA practice and for avoiding its further erosion.Targeted research efforts are needed to show possible ways out of the existing swamp of HRA inconsistencies and inadequacies. These would also deal with a thorough reconsideration and reevaluation of the applicability of toxicological or epidemiological approaches to HRA in specific situations [95,96,97,98]. Transparent addressing of the pros and cons of both approaches, and especially their uncertainties to improve the fitness for purpose, trustworthiness, confidence in, and credibility of the HRA process and its results, is inevitable.The idea and a need for distinguishing between facts (science) and values in planning and decision-making, despite being an old and repeatable subject of discussion [72,73,76,99,100], have either not been implemented, or they have, but with no better success in convincing stakeholders than the assessments that were missing them. An exploration of this with targeted research would be beneficial and interesting.

### 4.5. Limitations of the Study

The survey was limited in scope and depth. It focused on HRA elements in the decision-making context and not on a broad understanding of all RA principles and elements. The main limitations of the survey were the selection and representativeness of the target population groups and the response rate (between 24% and 58% in all groups). Higher response rates from the first three groups could be related to the authors’ and responders’ involvement in the same activities. Non-probabilistic judgmental sampling limited the use of advanced (probabilistic) statistical methods, which can limit the inference of our findings to larger populations (e.g., groups of professionals in the HRA, HIA, or decision analysis areas). While the formulation of certain questions and response options differed slightly between the two questionnaires, their meaning stayed the same. Therefore, this was not expected to contribute considerably to uncertainties when comparing the results between the groups. The largest proportion of the responders from the first three groups had a background in research, while the backgrounds in public health or economy were not represented in a comparable manner. More than half of all respondents in the first three groups declared having previous experience or involvement in decision-making cases that required the assessment of health risks or impacts. However, despite declaring this, our findings suggest that the respondents were not yet involved or lacked experience in risk-informed decision-making. Actual decision-makers who had been using the results of the HRA were not represented in the survey as much as the other groups were. Despite our efforts to involve professionals with knowledge and practical experience in risk and decision analysis, the findings of this study cannot represent a broad understanding of the survey topics. In addition, the study may have missed important HRA elements, although the respondents identified none. In-depth interviews with the majority of survey participants would be necessary to obtain deeper and more detailed insights into their understanding of the topics covered by the survey, which remains a topic for future research efforts in this arena. Nevertheless, the findings of our study provide valuable and rare insights into the understanding of links between HRA results and decision-making.

## 5. Conclusions

The paper addressed recent alerts on inadequate understandings and practices of HRA with a focus on the pressing erosion of confidence in the risk informing value of HRA in public health decision-making. Based on the survey results among different stakeholders involved in risk assessment and decision-making, which show inconsistent distribution of the responses, we highlight the need for procedural opportunities to improve the overall understanding and the interaction between HRA and environmental and public health decision-making. It is vital that clarification of the assessment and decision contexts among all relevant and involved stakeholders is done at the beginning of the HRA process. Additionally, a decision follow-up step is needed at the end of the process to evaluate the implementation of decisions and identify the actual success and benefits of the HRA. While more targeted research on various foundational and procedural RA issues is essential to show potential ways for dealing with them, the immediate effective consideration of the assessment context in all steps of HRA can already contribute to improvements in risk-informed decision-making in environmental health, public health, and beyond.

## Figures and Tables

**Figure 1 ijerph-19-04200-f001:**
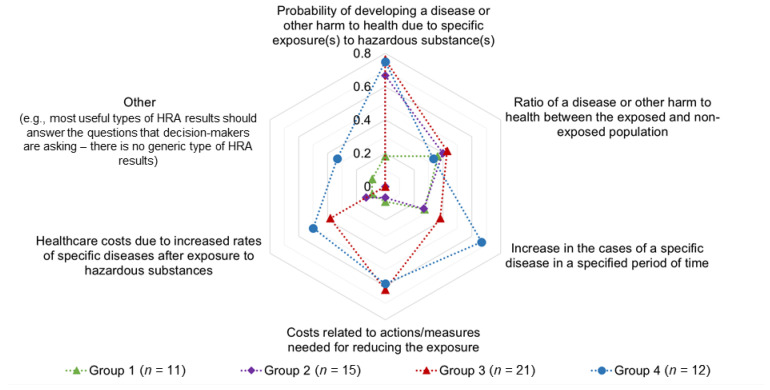
Most useful types of HRA results for decision-making—proportions of responses from the four groups.

**Figure 2 ijerph-19-04200-f002:**
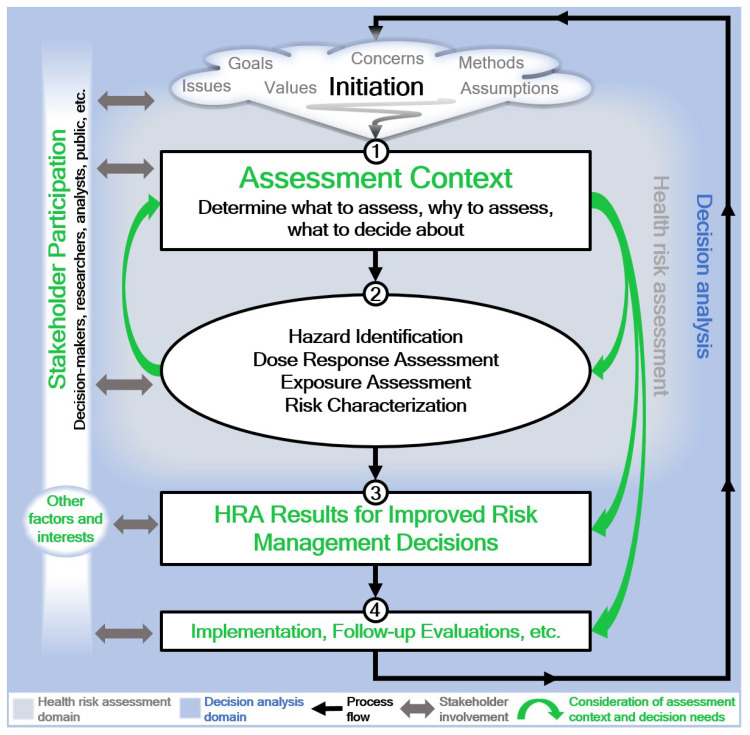
Opportunities for improving the utility of HRA.

**Table 1 ijerph-19-04200-t001:** Selected HRA concepts and comments on recognized issues.

Concept	Comments
Terminology and narrative	Interdisciplinary communication and collaboration are crucial in health risk assessment (HRA). Clear, consistent, and efficient terminology and narratives among all involved in HRA are essential. Experienced scientists with deep knowledge should be willing to patiently explain terms and definitions to less experienced ones.
Probability (uncertainty) and “HRA for chemicals”	The core of HRA is the probability (likelihood or frequency) of exposures and consequent health impacts. Contact with hazardous substances during human activities/habits and the physiological responses to these contacts (intakes) are subjects of probability (uncertainty), while the properties of hazardous substances are not subjects of probability but are deterministic (probability equal to 1). It is poor science to apply probability to deterministic parameters (substances and their properties) and, consequently, to calculate risk for them.
Hazard vs. Risk	These two concepts are too often interchanged. “Renaming” hazard into risk seems the easiest way to avoid probabilistic risk issues and related transparent calculations. Such reasoning has become widely adopted, particularly in using the hazard index and/or the risk characterization ratio as measures of risk.
No exposure no risk	This concept is clear, yet exposures are unknown, uncertain, and/or unjustified in many attempts to conduct HRA. “Inventing” exposure by applying unjustified assumptions, especially for wider populations, such as at national or regional levels, is inappropriate, particularly in the context of risk management. Such praxis leads to misinforming risk management. Consequently, inappropriate, non-justified societal decisions can be made. Invented, improperly justified exposures are inappropriate, even for teaching and training purposes. Trainees misuse training examples and exercises in their daily work, which allows for the wide dissemination of erroneous concepts.
HRA leading to or led by risk management?	The main reason for wanting to assess risks is to manage them by either reducing or removing their causes or the consequences, or both. Management decisions often involve balancing the advantages and disadvantages of the environment, human health, and the consequences for other social benefits of different options. This complex situation has led to the need for comprehensive decision analysis and should emphasize how the management context and criteria can, or indeed should, influence the HRA context.
Fitness for purpose	HRA performed without a clear purpose cannot provide clear information and scientific basis for informing actions that aim at specific improvements of health in a selected population. The purpose should reflect the expectations of the users of HRA results, which, in turn, influence all other elements of the HRA methodology and process.

**Table 2 ijerph-19-04200-t002:** Survey information.

	Target Group	Area of Work or Interest	Time Period	Level	Size	Responses
1.	Participants of the CRP V3-1722 ^1^ workshop	Administration, economy, public health, research	November 2019 to December 2019	National(Slovenia)	19	11
2.	Researchers involved in the NEUROSOME project ^2^	Research	December 2019 to June 2020	Regional(Europe)	29	15
3.	Participants in the “Environmental Health Risk: Analysis and Applications” educational activities ^3^	Administration, economy, public health, research	March 2020	Regional(United States)	38	21
4.	Established risk analysis and decision analysis professionals	Administration, economy, public health, research	November to December 2020	Global	49	12

^1^ Project title: Attempt at interpretation of biomonitoring results in connection with environmental pollution monitoring data, with the emphasis on air pollution and assessment of potential impacts of these pollutants on the health of inhabitants”; funded by the Slovenian Research Agency. ^2^ Principal investigators, early-stage researchers, and other researchers involved in the NEUROSOME project (https://www.neurosome.eu/, accessed 29 March 2022). ^3^ Organized by the Harvard T.H. Chan School of Public Health, 9–12 March 2020.

## Data Availability

Not applicable.

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
