# Peer review of "Practical Opportunities to Improve the Impact of Health Risk Assessment on Environmental and Public Health Decisions"

_ijerph, 2022, doi:10.3390/ijerph19074200_

Round 1

Reviewer 1 Report

  1. The calculation of the questionnaire score is based on a five-point Likert scale, and there are only three options. Is it intentional to increase the gap between the values of each option?
  2. The professional field distribution of the first three groups of samples is not concentrated, which may cause dispersion errors.
  3. The option of "I don't know" in some questions is counted as 0 points, and is included in the calculation of the average, which will underestimate the importance of the option.
  4. Q12. The mean of the first three groups to fill in the questionnaire does not match the actual calculation result.
  5. Answers to open-ended questions are not categorized or analyzed to explain the meaning of open-ended answers.

Reviewer 2 Report

Thanks the author provide a very clearly paper. I only have one minor suggestion. 

The introduction part is very clearly, however, it is too long. It would be better If the author can cut some sentences.

Reviewer 3 Report

The manuscript titled “Practical opportunities to improve the impact of health risk assessment on environmental and public health decisions” deals with a very interesting topic. The health risk assessment is important in public health decisions-making but this role is not adequately understood. Based on the survey results among different stakeholders involved in risk assessment and decision-making, the authors highlight the need for procedural opportunities to improve the overall understanding and the interaction between HRA and environmental and public health decision-making.

The paper is well-written and, although it has limitations, these have been clearly reported by the authors.

However, I believe that an explanation of the statistical methods of analysis should be included.

Reviewer 4 Report

The paper addresses the inadequate understandings and practices of health risk assessments (HRA) mainly due to the incorrect use of the terms hazard and risk. This misunderstanding is further supported by a survey, which included different stakeholders involved in risk assessment and decision-making, which show inconsistent distribution of the responses.

There is no doubt that a proper use of hazard and risk, its consequences for HRA and HIA and the procedure how to assess them is warranted so that a publication to address this problem may improve the situation. Since I assume that IJERPH is read by trained hazard- and risk-assessors who are well aware of the necessary elements of risk assessment I wonder if this journal is the right medium.

For publication in IJRPH the lengthy and confusing manuscript needs intensive revision.

In the Introduction the obvious problem should be addressed and its consequences for HRA and HIA, which will be further documented by a survey.

The many examples about the consequences of inappropriate use of the terminology spread all over the manuscript should be reduced.

Before their statement on glyphosate the authors better rely on the scientific literature and the conclusions of competent authorities rather than on US court decisions. Even ECHA concluded that glyphosate is not carcinogenic.

In the Discussion the outcome of the survey may be interpreted. Recommendations to improve the situation need to be based on an appropriate definition of hazard and risk, which is not provided so far.

Round 2

Reviewer 1 Report

The author's reply and related supplementary explanations are very detailed and clear.
I have no further comments.

Reviewer 4 Report

The authors have carefully considered the author's comments which improved readability  and focused the general message to differentiate between hazrd and risk.

I have no further comments.